# Effect of External Feedback on Bimanual Coordination Control in Patients with Parkinson’s Disease

**DOI:** 10.3390/ijerph182010927

**Published:** 2021-10-18

**Authors:** Eonho Kim, Chang-Ha Im, Yong-Gwan Song

**Affiliations:** 1Department of Physical Education, Dongguk University, Seoul 04620, Korea; eonkim@dongguk.edu; 2Department of Physical Education, Korea University, Seoul 02841, Korea; 3Department of Marine Sports, Pukyong National University, Busan 48513, Korea; 4Department of Marine Design Convergence Engineering, Pukyong National University, Busan 48513, Korea

**Keywords:** bimanual coordination, Parkinson’s disease, basal ganglia, movement disorders, information processing

## Abstract

Bimanual coordination control requires task-specific control of the spatial and temporal characteristics of the coupling of both upper limbs. The present study examined the effects of external feedback (i.e., auditory signal) on bimanual coordination movement during patients with Parkinson’s disease (PD). Twelve PD patients in advanced stages and 12 early stages of untreated PD patients, and 12 age-matched normal adults were instructed to perform bimanual coordination control using preference (1 Hz) and fast (1.75 Hz) speeds with metronome auditory cue. The results demonstrated that the advanced PD patients showed reduced synchronized bimanual coordination control during the anti-phase movement compared with other two groups. Moreover, the decreased movement accuracy was exhibited not only at the preference speed, but also more particularly at the fast speed with anti-phase rather than in-phase movement. This suggests that PD results in impairments in scaling the bimanual movement speed and amplitude of limb, and these deficits were more pronounced as a function of movement control speed. Overall, the current data provide evidence of the pathophysiology of the basal ganglia on the bimanual coordination movement.

## 1. Introduction

Major symptoms of motor disorders due to Parkinson’s disease (PD) include persistent resting tremor of the limbs, rigidity of the muscles during passive movement, posture instability, and bradykinesia with slowness of movement. Many voluntary movements such as bimanual coordination require the simultaneous integration of movements of the two upper limbs with task-specific control of the spatial and temporal characteristics [1].

Movement disorders lead to notable difficulties in the coordination of movements that require synergistic interactions between muscles or limbs. Coordination requires a complex process of neural integration both at the high and low level, and takes place temporally and spatially [1]. A typical example is bimanual coordination, comprising in-phase movements with both hands, with the right hand moving clockwise, and the left hand counterclockwise (symmetrically). In bimanual, anti-phase movements, both hands simultaneously move either clockwise or counterclockwise (asymmetrical). Asymmetrical bimanual movements, in which homologous muscle groups are activated alternately, are far more difficult to perform. When performing bimanual movement tasks, patients with PD and healthy subjects prefer to have the same temporal structure (i.e., symmetrical bimanual movements) of motions for both arms [2,3], and symmetrically coordinated motions in the same direction are more easily performed [4].

In fact, patients with PD show more unstable movements when performing cooperative tasks compared with normal people, and such unstable movements occur more prominently in anti-phase coordination tasks where the coordination pattern is more complex than simple in-phase movements [4,5]. In other words, patients with PD have difficulty in performing properly coordinated actions according to the imposed task conditions; as the task complexity increases, such as with anti-phase movements, disability in the spatiotemporal control of coordinated motion becomes more prominent [5,6]. This means that the central nervous system prefers to have the same structure of motion when performing cooperative movements [2,3]. In other words, the symmetrical coordination associated with two-arm motions in the same direction is relatively well performed compared with asymmetrical coordination in opposite directions [4,7,8].

Patients with PD lack coordination capabilities and dysfunction in anti-phase tasks is more noticeable as they require more attention than in-phase tasks. This phenomenon becomes clearer as the task complexity and information processing requirements increase. Performing in-phase motion is relatively easy because the two limbs are moving in the same direction. On the other hand, when generating an anti-phase coordination pattern, since the two limbs must be controlled in different directions, the perception and information processing requirements increase and the performance of the motion decreases. Patients with PD however, tend to rely heavily on externally provided motion signals or feedback in motor performance, alleviating difficulties in movements when auditory or visual signals are provided. This is supported by some studies showing that external signals (visual and auditory) lead to improvement in movements’ performance when performing gait [9] and sequential movements [10,11]. On the other hand, in other studies, the effectiveness of external feedback has not been de novo demonstrated, as the motor performance of PD varies depending on the stimulus or modality of the signal provided [6,12]. One possible explanation for speed-dependent (auditory signals) deficits is that the increase in execution demands imposed by an additional speed requirement of the task may accentuate the processing demands for the performance of complex bimanual movements more for those with advanced PD [7,13]. In conclusion, it is true that external signals help to promote movement performance, but controversy still exists as to how these stimuli are expressed depending on task difficulty (e.g., type of coordination) and type of task, and how useful they are for patients with PD.

This controversy can be thought of in two ways. First, previous studies have focused on patients with moderate-to-severe disability after the disease has progressed. When these patients have a long-term history of more than 5 years of medication use, the medicinal effects diminish, resulting in motor complications, such as dyskinesia, in which the limb is twisted and does not move as intended, as side effects. Drug resistance and the resulting motor complications may reduce the effects of external signals. In addition, few studies have been performed to determine how the progression of the disease or the adverse effects of drugs affect the presence or absence of external signals related to the alleviation of PD. Therefore, it is difficult to discover the mechanisms of improved movement performance in patients with PD by providing external signals as well as establishing an intervention strategy for rehabilitation. Secondly, previous studies have examined the effectiveness of extrinsic signals, mainly on discontinuous motion with a clear start and end.

Actually, bimanual coordination function in PD can be more disturbed with advancing disease stage [14]. Some studies have shown contradictory results in bimanual coordination movements depending on the disease stage or medication on and off. Therefore, the provision of external feedback may have an effect on the temporary task performance during drawing movements on a digitizer tablet [15], but the effect on other tasks with the tracking task to a reasonable level of temporal and spatial accuracy can be relatively reduced [7]. Therefore, it is necessary to confirm whether external signals can improve the movement performance of patients with PD through various movements’ tasks. In particular, no study has yet been reported on whether patients with PD are affected in consecutive motor tasks with external signals.

The present study aimed to investigate the difference in the ability to coordinate the two arms depending on whether the external signal was provided during the execution of the two-arm coordination movement, in de-novo patients who had been diagnosed with PD but had not received any medication yet, and patients with PD who had been taking medication for 5 years or more. We hypothesized that external feedback would help improve the performance of bimanual coordination movement in patients with PD. It was also assumed that this hypothesis would be more effective in the in-phase bimanual coordination task than in the anti-phase bimanual coordination task.

## 2. Materials and Methods

Twelve healthy controls (average age: 64.4 ± 5.2 years), 12 patients (average age: 65.9 ± 7.7 years) who were diagnosed with de novo PD and had no experience of drug treatment, and 12 patients with moderate PD (average age: 66.9 ± 6.9 years) participated in the study. They were all diagnosed by neurologists; the severity of PD was assessed using the UPDRS (Unified Parkinson’s Disease Rating Scale), an index to assess motor impairment and impairment due to PD. All participants were checked for cognitive abnormalities using the mini mental state examination (MMSE); all participants had a score of 25 or higher. The maximum MMSE score is 30 points. A score of 20 to 24 suggests mild dementia, 13 to 20 suggests moderate dementia, and lower than 12 indicates severe dementia. Additionally, all participants had no hearing problems.

The experiment used a two-arm coordinated horizontal motion system. Participants performed two-arm coordination movements in which they sat on a chair centered on the midline of the subject’s body, held two levers fixed on the table, and performed a two-arm coordination movements that repeated horizontal extension and flexion movements. In this study, the in-phase movement conditions consisted of simultaneous movements of both hands either towards (inwards) or away (outwards) from the body midline. The anti-phase movement conditions consisted of simultaneous movements of both hands either to the right (clockwise) or to the left (counterclockwise). Participants performed 15-s arm-to-arm movements consistent with the two task types (in-phase, anti-phase) shown on the computer monitor. The in-phase movement caused both muscles of both arms to contract and relax simultaneously (0 ° phase difference), and the anti-phase movement repeatedly crossed the same kind of muscle groups with repeated flexion and extension (90° phase difference). The two-arm coordination movement requires approximately 80° of motion on both the right and left sides. The left and right displacements were displayed on the vertical and horizontal coordinates of the computer screen, respectively. The analog displacement information of the performed motion was converted to digital displacement information through an A/D converter and stored on the computer.

The experimental task involved the performance of in-phase and anti-phase coordinated movements presented on the monitor according to the speed of an auditory signal. The auditory signals were presented via speakers located to the left and right of the monitor at one of two rates (1 Hz and 1.75 Hz) using computer-programmed metronomes. Participants were instructed to adjust the period of movement according to the presented auditory signal. The accuracy of the two-arm coordination was measured by repeating the extension and flexion of both arms to 18 times in total for 10 s. The experiment was divided into two movements’ mode sections, and each practice section was divided into 12 movement. The first three trials comprised an in-phase coordination task, and the next three trials comprised an anti-phase coordination task. For each task, the two rate conditions were performed three times, and the order was randomized for each participant. A one-minute break was given between tasks, and a five-minute break between the two practice intervals was also included. For half of the movement section, auditory signals were sent for the first 5 s and not for the remaining 10 s. The participant was instructed to keep performing the coordinated movement at the given rate and finish it to the end. Prior to the experiment, all participants performed a maximum of five bimanual coordination movement to familiarize themselves with each coordinated movement.

The accuracy of the performance of the two-arm coordination task was calculated using the relative phase (RP). The relative phase is a variable that can characterize coordinated activities between segments or limbs. It was used as a means of identifying the stability of the coordination in body parts that are connected to one collective unit rather than moving independently. The relative phase represents the phase relation between the mutual movements of the two joints and their angular velocities and is a variable that expresses the temporal and spatial coordination patterns of the body components [16]. The individual performance characteristics of each limb during the coordinated operation were measured using motion speed based on frequency analysis, operating distance between maximum flexion and extension, and the continuity of motion in the performance of a given task. The movement harmonicity (H) used by Guiard [17] was calculated as a measure of continuous movement without hesitation or stoppage during the operation. The H-value is obtained as the ratio of the absolute value of the minimum acceleration to the maximum acceleration within one cycle of each movement. The H-value is close to 1 if it is performed continuously without any hesitancy or interruption in the execution of a given task and approaches 0 when the operation is stopped or intermittently performed. The operating period was calculated using a Fourier transform, and the working distance was calculated as the absolute difference between the maximum and minimum values of the elbow joint angle.

For the analysis of the quantified data, the mean and standard deviation of each error value were calculated using the SAS 9.1 statistical program. To verify the measurement variables in each coordination condition and rate condition, two-way analysis of variance with repeated measures on the last factor (ANOVA) between the groups of patients (de novo Parkinson’s disease, moderate-level Parkinson’s disease, healthy control group- level 3) × external signals (level 2). Duncan’s multiple-range test was used for post hoc comparisons and simple main effects analysis was used if the interaction effect was significant. The statistical significance level was set at 0.05.

## 3. Results

### 3.1. Control of Bimanual Coordination Movements

#### 3.1.1. In-Phase

The performance of the cooperative task according to the given coordination condition and rate condition was analyzed based on the average error of the relative phase (Figure 1). The result shows that there were no significant difference among the groups in the relative phase mean error values in the preference rate condition of in-phase coordination and that there was not a significant difference with respect to the presence or absence of external signals [F(1,13) = 2.54, *p* = 0.11]. In other words, the mean error value of the relative phase increased in all three groups in the absence of external signals. The groups × external signal conditions interaction effect was not significant. In the in-phase fast rate condition (1.75 Hz), the result is similar to the preference speed condition (1 Hz). There were no significant difference in the relative phase mean error among the groups, but there was a significant difference in whether the external signals were present [F(1,13) = 4.02, *p* = 0.01]. Likewise, in the absence of external signals, the mean error of the relative phase increased in all three groups. The interaction effect among the groups × external signals was not significant.

#### 3.1.2. Anti-Phase

In the preference condition (1 Hz) for anti-phase coordination (Figure 1), there were no significant difference among the groups in the relative phase mean error value, but the main effect according to the external signal condition was significant [F(1,13) = 3.18, *p* = 0.05]. The group × external signal conditions interaction effect did not show a significant difference [F(2,16) = 2.55, *p* = 0.08]. In the anti-phase fast rate condition (1.75 Hz), the result is similar to the preference rate condition (1 Hz). There were no significant difference in the relative phase mean error among the groups, and there were no significant difference when the external signals were present and when they were not [F(1,13) = 3.11, *p* = 0.11]. The group × external signal condition interaction effect was not significant but was close to the threshold value of significance [F(2,16) = 2.74, *p* = 0.09].

### 3.2. Control of Bimanual Coordination Amplitude

#### 3.2.1. In-Phase

As a result of analyzing the effect of control ability on the target distance (e.g., 80° joint angle) given in the task (Figure 2), there was a significant difference in the target distance among the groups in the slow rate of in-phase movement (1 Hz) [F(2,22) = 5.24, *p* = 0.01]. Post-hoc analysis showed that the healthy control group had a larger working distance than the two Parkinson’s disease groups, with both Parkinson’s patient groups exhibit a lower working distance than the target distance, with no significant difference between them. However, the presence or absence of external signals did not cause a significant difference in operating distance. The groups × external signal conditions interaction was not significant. Even in the fast condition (1.75 Hz), there was a significant difference in the movement distance among the groups. [F(2,22) = 6.14, *p* = 0.01]. Post hoc analysis showed that the normal group was closer to the target distance than the two PD groups and the two PD groups showed a significantly a lower range of motion than in the slower rate condition. The presence or absence of external signals did not lead to a significant difference in operating distance. The interaction effect among the groups × external signal conditions was not significant [F(2,22) = 2.15, *p* = 0.11]. Post hoc analysis results indicated that the working distance of the de novo PD group was significantly reduced when the external signal was not provided compared with the healthy control group. In patients with moderate PD who have a history of taking medication, the decrease in the distance of motion was more pronounced in the absence of external signals than in patients with de novo PD (Figure 2).

#### 3.2.2. Anti-Phase

In the preference rate condition of anti-phase coordination (1 Hz), there was a significant difference in distance among the groups [F(2,22) = 5.14, *p* = 0.01]. Post-hoc analysis showed that the healthy control group had a significantly larger range of motion than the other two PD patient groups. The group of patients with severe PD who had taken medication showed a further reduced range of motion compared with the de novo PD group. The presence or absence of external signals did not cause a significant difference in the working distance, but the interaction effect among the groups × external signal conditions showed a significant difference [F(2,22) = 5.89, *p* = 0.01]. The results of the follow-up analysis showed that compared with the healthy control group, the motion distance of the de novo PD group was significantly reduced when the external signal was not provided; this reduction was more pronounced in the patient group with moderate PD who had taken medication (Figure 2). A similar pattern to the preference rate condition was observed in the fast rate condition (1.75 Hz). There was a significant difference in the motion distance among the groups [F(2,22) = 9.21, *p* < 0.01]. Post hoc analysis showed that the healthy control group had a slightly higher working distance than the target distance, but the two PD groups exhibited a significantly lower working distance than the target distance. There were no significant difference in the presence or absence of an external signal, but the group × external signal condition interaction was significantly different [F(2,22) = 3.56, *p* = 0.05]. Post hoc analysis showed that motion distance in the de novo PD group was significantly lower than that in the healthy control group when an external signal was not provided; the decrease in motion distance in the severe PD group who took medications, was more pronounced than that in the de novo PD group.

### 3.3. Control Bimanual Coordination Harmonicity

#### 3.3.1. In-Phase

As a result of analyzing the ability to perform a given task consecutively without hesitation or interruption (Figure 3), there was a significant difference in the continuity values among the groups in the preference condition (1 Hz) [F(2,22) = 24.14, *p* < 0.01]. Post hoc analysis showed that the group of patients with severe PD who had taken the drug showed significantly lower continuity of motion than the other two groups. However, the presence or absence of external signals did not cause a significant difference in the continuity of movement. The group × external signal conditions interaction was not significant. In the in-phase fast rate condition (1.75 Hz), the overall motion continuity was higher than that in the preference rate condition. However, statistical analysis showed no significant difference in the continuity of motion among the groups and external signal conditions. That is, in the fast rate condition, hesitancy or discontinuity of motion decreased due to the increase in the speed of motion, regardless of whether the external signal was provided, in all groups.

#### 3.3.2. Anti-Phase

Similar to the in-phase condition, the preference rate condition of anti-phase coordination (0.75 Hz) showed a significant difference in continuity among the groups [F(2,22) = 25.18, *p* < 0.01]. Post hoc analysis showed that the movement continuity of the patient group with severe PD who had been taking medication was significantly lower than that of the other two groups. The interaction effect among the external signal condition [F(1,13) = 10.13, *p* = 0.01] and group × external signal condition showed a significant difference [F(2,22) = 8.24, *p* = 0.01]. Post hoc analysis showed that there was a significant difference in the continuity of motion in the severe PD group, and the de novo PD and healthy control groups, dependent on the presence or absence of external signals. The continuity of operation increased compared to the slow rate condition in the fast rate condition (1.75 Hz) for the anti-phase task. However, there were no significant difference in the continuity of motion among the groups or external signal conditions. That is, in the fast rate condition, continuity of operation is high, regardless of whether the external signals are provided, in all groups.

## 4. Discussion

The purpose of this study was to investigate the effects of external feedback in patients with de novo and moderate PD in the production of bimanual coordination. The results demonstrate that patients with PD (de novo PD, PD) showed worse bimanual coordination patterns in the anti-phase than in-phase movements. PD patients showed a more accurate and stable coordination movement at the preference speed than fast speed. However, compared to with healthy controls (NC), patients with PD (de novo PD, PD) showed more significant movement disturbances when required to perform movement at a higher speed than when performing slow movements, and these abnormal movements were more evident in the anti-phase coordination pattern than in the in-phase coordination pattern.

The external signal is known to act as a temporal or spatial stimulus provided from the outside during movement to facilitate the initiation of movement and to assist its continuous performance. In this study, external signal (auditory cues) at the preference speed did in result more accurate bimanual coordination movement. This finding suggests that movement speed (preference or fast) seems to have a greater impact on the performance of bimanual coordination tasks than movement mode (in-phase or anti-phase). These results are consistent with the fact that PD patients experience more difficulty than healthy controls in performing the required movement at higher frequency (1.75 Hz) compared to lower frequency (1 Hz) [6,15]. In contrast, other studies have suggested that there is no influence of movement speed on bimanual coordination task performance [18,19,20]. These results concerning the influence of different external cue types on movement performance are somewhat consistent with previous studies.

One explanation for the benefit of external feedback may allow for kinesthetic focus which has been shown to increase brain activation in PD patients. When a stimulus or cue related to the movement is provided from the outside, it is believed that this utilizes a neuronal circuit connected to the relatively undamaged area of the cerebral premotor area and does not use the main neuronal circuit that reaches the complementary motor area through the basal ganglia [12]. That is, the activation of other neurotransmission pathways by the provision of external signals can be used as a compensatory strategy for improving bimanual coordination movement’s performance by bypassing the functionally impaired basal ganglia neuron due to PD [7,12]. Thus, external feedback with respect to preference speed should help to improve bimanual coordination movement in patients with PD.

Previous studies have investigated patients with moderate-to-severe PD, while this study included patients with early symptoms of PD (de novo). Moreover, most studies involved PD patients receiving optimally balanced Parkinsonian medication and investigated the ‘on’ medication condition. To date, little is known about the effects of dopaminergic drugs on bimanual coordination movement. The present study investigated the effects of dopaminergic medication on bimanual coordination movement with external feedback speeds. Unlike patients with de novo PD, patients with moderate PD exhibited greater difficulty in performing bimanual coordination movements with anti-phase mode rather than in-phase mode. In other words, in the anti-phase coordination task, the provision of external signals did not induce a positive change in the ability to perform two-arm coordinated movements in patients with moderate PD. Considering the fact that dopaminergic treatment improves SMA (supplementary motor area) activity in PD patients performing coordinated movements [21], at least some effect should be expected in patients with PD (de novo or advanced). These results indicate that if the complexity or difficulty of the task increases, such as with anti-phase movement in which the two arms are used to perform asymmetrical movements in a coordinated task, the provided external signal does not have a significant effect on performance in patients with moderate PD who have taken drug treatment for more than 5 years.

Almeida et al. [6] reported that patients with PD exhibited freezing in anti-phase coordination and that the provision of external auditory signals did not improve coordinated performance. However, in patients with de novo PD, the accuracy of coordinated movements was reduced to the level of patients with moderate PD when external signals were not provided. In other words, the provision of external signals is an important factor in the performance of patients with de novo PD. Moreover, patients with de novo PD had difficulty performing when intentionally planning and performing an operation without the provision of external signals while performing an anti-phase coordination task where the complexity of coordination is relatively high. These results indicate that the decrease in performance of the task itself is due to the inherent movement disorder caused by neurodegeneration of the basal ganglia in the cerebrum, rather than by the progression of the disease or adverse effects on movements due to drug treatment [7,15].

Many studies have demonstrated that anti-phase coordination movements are not as stable nor as accurate as in-phase movements [5,7,8,15]. In the study, PD patients were found to have more difficulty at performing bimanual coordination movement than NC. Compared with NC, this difference in PD patients was more pronounced in the anti-phase bimanual coordination movement than in-phase bimanual coordination movement. Patients with PD (de novo PD, PD) all had greater difficulty maintaining bimanual coordination movement performance at a fast speed than at the preference speed. In general, when spatial accuracy or movement speed demands are high, there is a greater cognitive information processing and cautionary capacity. Patients with PD are more likely to exhibit motor deficits when performing tasks with higher difficulty levels than slower tasks with lower task complexity. In particular, bradykinesia in patients with PD is more prominent in tasks requiring spatial accuracy and speed of movement [22,23], suggesting that bradykinesia prevents patients from properly bimanual coordination movement performing in fast speed. As a result, the ability to maintain the movement speed by regulating the cyclic movement of both arms is not significantly affected by the presence or absence of external signals. That is, it is considered that the presentation of external signals does not alleviate bradykinetic symptoms, which is the primary and inherent abnormal movement symptom caused by neuronal damage of the basal ganglia and is also not significantly related to the degree of disease progression. However, with bradykinesia, patients with PD showed markedly decreased motion control (undershooting) as the required speed of motion and complexity of coordination increased, and this movement disorder was more pronounced within the moderate PD group in which the disease was more severe.

As a result of the continuity of movement patterns according to the external signals, patients with de novo PD performed similarly to healthy people, whereas patients with moderate PD exhibited a short-lived behavioral pattern in which hesitancy or discontinuity of motion occurred. Furthermore, discontinuous motion is more pronounced at slower speeds than at faster speeds. In other words, when asked to perform a given task somewhat slowly, patients with moderate PD showed difficulty in performing the task flexibly and continuously. This result suggests that the decrease in the continuous performance of the movement is not an essential feature caused by the neural imbalance of the basal ganglia, but rather is a movement disorder caused by the progression of the disease or side effects due to long-term use of the drug. In previous studies, patients with PD reported difficulty in precisely controlling motion using internal and external sensory feedback [24,25]. This suggests that the ability to control the accuracy of motion continuously using sensory information was compromised as PD progressed and the task could not be performed smoothly and continuously. However, since the external signal only affected the moderate PD group, the use of a normal bypass neuron seems to alleviate dysfunction related to the perception of sensory information associated with the performance of a movement or sensory-motor integration.

## 5. Conclusions

In conclusion, the results of this study demonstrate that bimanual coordination movement dysfunction is a very early motor impairment in PD. However, in the study, external feedback can lead to improved bimanual coordination movement in the preference speed than fast speed. Moreover, this phenomenon shows that in-phase bimanual coordination movement is more useful than anti-phase bimanual coordination movement. Future research investigating timing and bimanual coordination in PD patients should include cognitive tasks and continue to examine different external feedback as an aid to perform complex coordination movement tasks.

## Figures and Tables

**Figure 1 ijerph-18-10927-f001:**
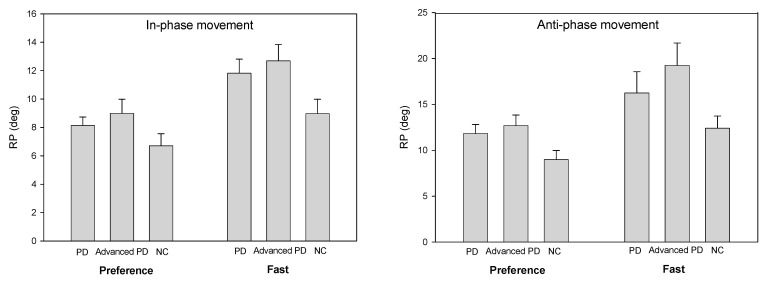
The groups average and the standard error of the relative phase (RP) as a external movement speeds.

**Figure 2 ijerph-18-10927-f002:**
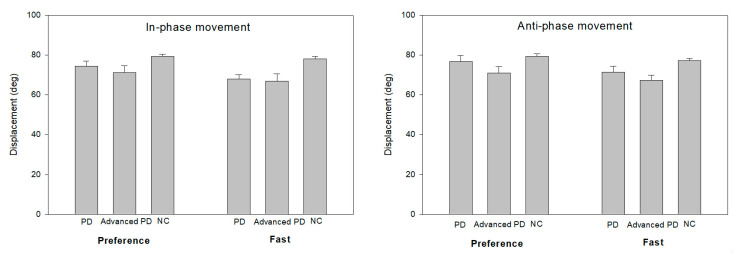
The groups average and the SE of the movement amplitude as a external movement speeds.

**Figure 3 ijerph-18-10927-f003:**
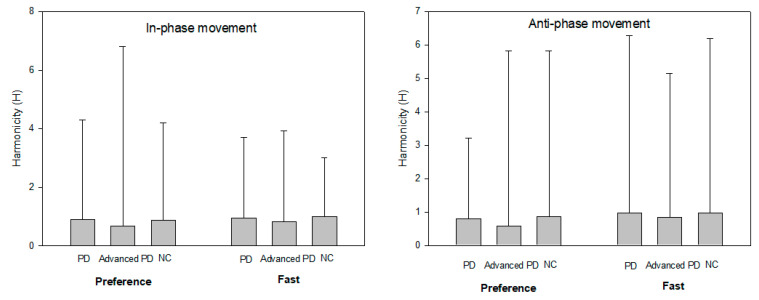
The group average and the SE of the harmonicity (H) as a external movement speeds.

## Data Availability

The data that support the findings of this study are available from the corresponding author upon reasonable request due to ethical and privacy restrictions.

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
