# Peer review of "Effect of External Feedback on Bimanual Coordination Control in Patients with Parkinson’s Disease"

_ijerph, 2021, doi:10.3390/ijerph182010927_

Round 1

Reviewer 1 Report

Thank you for the opportunity of reviewing this paper on this interesting topic, Effects of external Feeedback on Bimanual Coordination on Parkinson patients.

I have some suggestions for the authors in order to improve the quality of the manuscript.

I strongly recommend some English Editing. I have started correcting some of the grammar mistakes, but just stopped when I realized that there were too many of them.

Abstract: The abstract is well written and provides sufficient information to the readers.

Line 18: Please rephrase this sentence. It is to be expected that at the fastest speed, there would be less accuracy. The way the phrase is expressed right now, invites to think otherwise.

Keywords: I would recommend to include Parkinson in the keywords.

Introduction:

Line 30: “involving” “require”. Please check the grammar.

Line 35: I recommend some English editing. “the central nervous system prefers ….” I get the meaning, but a structuro like CNS is not a subject that prefers or not to do things.

Line 48: “in opposite directions”

Line 62: This is an interesting point. Please explain further in the text the nature of the stimuli and their different effect on patients.

Line 78 Please define further or set examples of Temporary exercise task and other exercise tasks.

I quite disagree with the focus set on medication and adverse effects, as Parkinson is a degenerative disease that will, due to its nature, diminish patients’ abilities with time. I mean, after 5 years of evolution, you cannot make the statement that adverse effects of medication are to be blamed for dyskinesia. Maybe you can comment something about this or include it as a limitation or alternative rationale.

Materials and Methods.

Did you make a sample size calculation for your study? If so, include it in the manuscript.

I assume that you also checked for auditive impairment. Specify that in the text, please.

Line 96: Explain what the score of 25 or higher means.

Line 100: Specify if the motion involved the shoulder joint, elbow joint or both. (I assume it was elbow motion, and it is clarified in line 146, but I still feel that it has to be explained earlier in the text)

Discussion:

I have some difficulty understanding your results. In line 262 you state:

. Regarding external signals, the success of coordinated actions of patients with PD was found to be better when not provided.

But in line 273:

If this external signal is provided during movement in a Parkinson's 272 patient, symptoms of the disease, such as slowed motion or reduced stride, are alleviated 273 [8, 11].

The explanation does not support your results.

Line 284 to 304. The whole paragraph is a bit confusing. Please, find a way to make it more understandable for the reader. Maybe a graphic?

Line 329 to 331. I feel I don’t quite understand what the authors mean by “t is not an essential feature caused by the neural imbalance of the basal ganglia but rather is a movement disorder caused by the progression of the disease” Does not the disease cause a progressive imbalance of the basal ganglia? Does not this neural imbalance get worse with time? If so, the sentence is not correct as it is written.

Conclusions:

Again, I am confused by the information provided in line 262.  You conclude that provision of external signals is helpful, but in that line “Regarding external signals, the success of coordinated actions of patients with PD was found to be better when not provided.” ,you state the opposite.

Overall, I find the manuscript interesting, but the authors need to work on some changes before it is suitable for publication.

Author Response

Dear Editor Adelina Platon and Reviewer

Thank you for your editorial decision letter on our manuscript submission. We and appreciate the care and insight given to the consideration of our manuscript.

In this letter, we explain how we addressed and solved each concern raised by you, Reviewer 1 and Reviewer 2. Please find our replies in the red font.

Reviewer #1

Line 18: Please rephrase this sentence. It is to be expected that at the fastest speed, there would be less accuracy. The way the phrase is expressed right now, invites to think otherwise.

→ We changed to “Moreover, the decreased movement accuracy exhibited not only at the preference speed, but also particularly at the fast speed with anti-phase than in-phase movement.”.

Keywords: I would recommend to include Parkinson in the keywords.

→ We added “Parkinson’s disease”.

Introduction:

Line 30: “involving” “require”. Please check the grammar.

→ We revised the grammar. And as your suggested, I reviewed and revised the manuscript by a native speaker. So, I modified such as “Major symptoms of motor disorders due to Parkinson's disease (PD) include persistent resting tremor of the limbs, rigidity of the muscles during passive movement, posture instability, and bradykinesia with slowness of movement. Many voluntary movements such as bimanual coordination requires the simultaneous integration of movements of the two upper limbs with task-specific control of the spatial and temporal characteristics

Line 35: I recommend some English editing. “the central nervous system prefers ….” I get the meaning, but a structuro like CNS is not a subject that prefers or not to do things.

→ We modified by the “A typical example is bimanual coordination, in-phase movements with both hands, the right hand moving clockwise, the left hand counterclockwise (symmetrical). In bimanual, anti-phase movements, both hands simultaneously move either clockwise or counterclockwise (asymmetrical). Asymmetrical bimanual movements, in which homologous muscle groups are activated alternately, are far more difficult to perform. When performing bimanual movement tasks, the patients with PD and healthy subjects prefers to have the same temporal structure (i.e., symmetrical bimanual movements) of motions for both arms [2,3], and symmetrically coordinated motions in the same direction are more easily performed [4]

Line 48: “in opposite directions”

→ We changed to it.

Line 62: This is an interesting point. Please explain further in the text the nature of the stimuli and their different effect on patients.

→ Thank you. That’s good point. We have provided evidence to support our opinion. “One possible explanation for speed-dependent (auditory signals) deficits is that the increase in execution demands imposed by an additional speed requirement of the task may accentuate the processing demands for the performance of complex bimanual movements more for those with advanced PD [8, 14]

Line 78 Please define further or set examples of Temporary exercise task and other exercise tasks.

I quite disagree with the focus set on medication and adverse effects, as Parkinson is a degenerative disease that will, due to its nature, diminish patients’ abilities with time. I mean, after 5 years of evolution, you cannot make the statement that adverse effects of medication are to be blamed for dyskinesia. Maybe you can comment something about this or include it as a limitation or alternative rationale.

→ Thank you. That’s good point. We revised the as follows “Actually, bimanual coordination function in PD can be more disturbed with advancing disease stage [15]. Some studies have shown contradictory results in bimanual coordination movements depending on the disease stage or medication on and off. Therefore, the provision of external feedback may have an effect on the temporary task performance during drawing movements on a digitizer tablet [16], but the effect on other tasks with the tracking task to a reasonable level of temporal and spatial accuracy can be relatively reduced [8]. Therefore, it is necessary to confirm whether external signals can improve the movement performance of patients with PD through various movements’ tasks. In particular, no study has yet been reported on whether patients with PD are affected in consecutive motor tasks with external signals

Materials and Methods.

Did you make a sample size calculation for your study? If so, include it in the manuscript.

→ Sorry, we didn’t calculate the sample size.

I assume that you also checked for auditive impairment. Specify that in the text, please.

Line 96: Explain what the score of 25 or higher means.

→ We added the such as “A score of 20 to 24 suggests mild dementia, 13 to 20 suggests moderate dementia, and less than 12 indicates severe dementia. Additionally, all participants had no hearing problems

Line 100: Specify if the motion involved the shoulder joint, elbow joint or both. (I assume it was elbow motion, and it is clarified in line 146, but I still feel that it has to be explained earlier in the text)

→ We added by the “In this study, the in-phase movement conditions consisted of simultaneous movements of both hands either towards (inwards) or away (outwards) from the body midline. The anti-phase movement conditions consisted of simultaneous movements of both hands either to the right (clockwise) or to the left (counterclockwise).

Discussion:

I have some difficulty understanding your results. In line 262 you state:

Regarding external signals, the success of coordinated actions of patients with PD was found to be better when not provided.

But in line 273:

If this external signal is provided during movement in a Parkinson's 272 patient, symptoms of the disease, such as slowed motion or reduced stride, are alleviated 273 [8, 11].

The explanation does not support your results.

Line 284 to 304. The whole paragraph is a bit confusing. Please, find a way to make it more understandable for the reader. Maybe a graphic?

Line 329 to 331. I feel I don’t quite understand what the authors mean by “t is not an essential feature caused by the neural imbalance of the basal ganglia but rather is a movement disorder caused by the progression of the disease” Does not the disease cause a progressive imbalance of the basal ganglia? Does not this neural imbalance get worse with time? If so, the sentence is not correct as it is written.

In the discussion session, we revised the manuscript based on our results. And, by adding related previous studies, we emphasized the research results, logic, and generalization. I would appreciate it if you could refer to the newly revised contexts in our manuscript.

Conclusions:

Again, I am confused by the information provided in line 262.  You conclude that provision of external signals is helpful, but in that line “Regarding external signals, the success of coordinated actions of patients with PD was found to be better when not provided.” ,you state the opposite.

→ We revised the contents according to the results. “In conclusion, the results of this study demonstrate that bimanual coordination movement dysfunction is a very early motor impairment in PD. However, in the study, external feedback can lead to improved bimanual coordination movement in the preference speed than fast speed. Moreover, this phenomenon shows that in-phase bimanual coordination movement is more useful than anti-phase bimanual coordination movement. Future research investigating timing and bimanual coordination in PD patients should include cognitive tasks and continue to examine different external feedback as an aid to perform complex coordination movement tasks.”

Reviewer 2 Report

Introduction

Paragraph 1 – needs references

Paragraph 3 – please consider indicating that it is more pronounced for people with PD, but we all have the same trend. In fact, it is easier to perform in-phase them anti-phase even for healthy people.

Paragraph 5 - needs references

Please consider including some hypothesis

Methods

The in-phase and anti-phase was only considered in 0º and 180º respectively? What about the degrees in between?

Results

Line 172 and 177 – Please consider changing “between” to “among”

Line 178 – Please consider removing the word “between”

Line 184 – Please consider removing “As a result”

Conclusions

Line 340 – Please consider rewrite the sentence

Author Response

Dear Editor Adelina Platon and Reviewer

Thank you for your editorial decision letter on our manuscript submission. We and appreciate the care and insight given to the consideration of our manuscript.

In this letter, we explain how we addressed and solved each concern raised by you, Reviewer 1 and Reviewer 2. Please find our replies in the red font.

Reviewer #2

Paragraph 1 – needs references

→ We wrote a reference. Thank you.

Paragraph 3 – please consider indicating that it is more pronounced for people with PD, but we all have the same trend. In fact, it is easier to perform in-phase them anti-phase even for healthy people.

→ That’s good point. We tried to increase the justification through additional explanations in the paragraph 2. The added content is as follows:

 “A typical example is bimanual coordination, in-phase movements with both hands, the right hand moving clockwise, the left hand counterclockwise (symmetrical). In bimanual, anti-phase movements, both hands simultaneously move either clockwise or counterclockwise (asymmetrical). Asymmetrical bimanual movements, in which homologous muscle groups are activated alternately, are far more difficult to perform.

Paragraph 5 - needs references

→ We changed to “Actually, bimanual coordination function in PD can be more disturbed with advancing disease stage [15]. Some studies have shown contradictory results in bimanual coordination movements depending on the disease stage or medication on and off. Therefore, the provision of external feedback may have an effect on the temporary task performance during drawing movements on a digitizer tablet [16], but the effect on other tasks with the tracking task to a reasonable level of temporal and spatial accuracy can be relatively reduced [8]. Therefore, it is necessary to confirm whether external signals can improve the movement performance of patients with PD through various movements’ tasks. In particular, no study has yet been reported on whether patients with PD are affected in consecutive motor tasks with external signals”.

Please consider including some hypothesis

 → Thank you. We added such as “We hypothesized that external feedback would help improve performing bimanual coordination movement in patients with PD. It was also assumed that this hypothesis would be more effective in the in-phase bimanual coordination task than in the anti-phase bimanual coordination task”

Methods

The in-phase and anti-phase was only considered in 0º and 180º respectively? What about the degrees in between?

 → That’s right. We consider in 0 º (in-phase) and 90 º (anti-phase). The 180 º is incorrectly written. I’m sorry

Results

Line 172 and 177 – Please consider changing “between” to “among”

→ We changed to it.

Line 178 – Please consider removing the word “between”

→ We remove to it.

Line 184 – Please consider removing “As a result”

 → We remove to it.

Conclusions

Line 340 – Please consider rewrite the sentence

→ We revised the contents according to the results. “In conclusion, the results of this study demonstrate that bimanual coordination movement dysfunction is a very early motor impairment in PD. However, in the study, external feedback can lead to improved bimanual coordination movement in the preference speed than fast speed. Moreover, this phenomenon shows that in-phase bimanual coordination movement is more useful than anti-phase bimanual coordination movement. Future research investigating timing and bimanual coordination in PD patients should include cognitive tasks and continue to examine different external feedback as an aid to perform complex coordination movement tasks.”

Round 2

Reviewer 1 Report

After carefully reading the paper I think the authors addressed all my concerns, and made a great effort improving the manuscript. It is really a relevant and interesting topic and the limitations of the study had been addressed adequately. The modifications in the discussion section have made it clearer and more in line with the results.